# Desertification–Scientific Versus Political Realities

**Geert Sterk [1,\*] and Jetse J. Stoorvogel [2]**

[1]  Department of Physical Geography, Faculty of Geosciences, Utrecht University, 3584 CB Utrecht, The Netherlands
[2]  Soil Geography and Landscape Group, Wageningen University, 6708 PB Wageningen, The Netherlands; jetse.stoorvogel@wur.nl
\*  Correspondence: g.sterk@uu.nl; Tel.: +31-30-2533051

**Abstract:** Desertification is defined as land degradation occurring in the global drylands. It is one of the global problems targeted under the Sustainable Development Goals (SDG 15). The aim of this article is to review the history of desertification and to evaluate the scientific evidence for desertification spread and severity. First quantitative estimates of the global extent and severity of desertification were dramatic and resulted in the establishment of the UN Convention to Combat Desertification (UNCCD) in 1994. UNCCD's task is to mitigate the negative impacts of desertification in drylands. Since the late 1990s, science has become increasingly critical towards the role of desertification in sustainable land use and food production. Many of the dramatic global assessments of desertification in the 1970s and 1980s were heavily criticized by scientists working in drylands. The used methodologies and the lack of ground-based evidence gave rise to critical reflections on desertification. Some even called desertification a myth. Later desertification assessments relied on remote sensing imagery and mapped vegetation changes in drylands. No examples of large areas completely degraded were found in the scientific literature. In science, desertification is now perceived as a local feature that certainly exists but is not as devastating as was earlier believed. However, the policy arena continues to stress the severity of the problem. Claims that millions of hectares of once productive land are annually lost due to desertification are regularly made. This highlights the disconnection between science and policy, and there is an urgent need for better dialogue in order to achieve SDG 15.

**Keywords:** desertification; land degradation; global assessment; drylands; Sahel

## 1. Introduction

Desertification is an environmental issue that has been and still is a topic of political and scientific debate. The term desertification refers to land degradation in the Earth's drylands, but has been interpreted in many different ways. In the 1970s, when the early satellite images became available to science and the general public, desertification was often related to the southward extension of the Sahara desert [1,2]. This perception appeared to be wrong, and since then a lot of debate has arisen about the definition of desertification, its causes, the severity, the global occurrence of desertification, and the impacts it has on the dryland populations. Despite the confusion and different interpretations of what desertification actually means, alarming numbers on the extent of desertification were published. Estimates of 50% and more of the drylands being affected to some extent by desertification were published [3,4], but these numbers were heavily criticized by scientists actually working in dryland environments [5].

Today, it is still unclear how big a problem desertification is for agriculture and food security. The United Nations through its Convention to Combat Desertification (UNCCD) claims that 12 million hectares of land are annually lost through desertification and drought, which is equivalent to a loss of

20 million tons of grain production [6]. Many similar statements are regularly made. For instance, the International Fund for Agricultural Development [7] claimed that:

> "The livelihoods of over 1.2 billion people inhabiting dryland areas in 110 countries are currently threatened by drought and desertification."

On the other hand, there are scientists who have claimed that the whole concept of desertification is nothing but a myth [8]. They claim that the evidence for the massive and unprecedented desertification is lacking. According to Behnke and Mortimore [9], the whole discussion about desertification has become disconnected between policy and science. While science seems to dispute the very existence and severity of desertification, the policy arena continues to claim that it is a serious threat for sustainable agriculture and food production in the global drylands. In fact, the UN has defined Sustainable Development Goal (SDG) 15 which includes combatting desertification as an important target for the sustainable use of terrestrial agro-ecosystems.

The aim of this article is to review the history of desertification research and evaluate the scientific evidence for desertification spread and severity. We attempted to review the most relevant peer-reviewed published journal articles, as well as the official international reports published by the main policy bodies such as UNCCD. The article concludes with a section on the scientific evidence of the degree and global spread of desertification, and the existing disparities between the scientific and policy domains.

## 2. History of Desertification

The use of the term desertification has been claimed to originally stem from the colonial French and British administrations in West Africa in the 1920s and 1930s [10]. The word desertification was not yet used, but foresters were worried about increasing drought and the creation of desert-like conditions on crop and grazing land due to overgrazing and uncontrolled burning. According to D'Odorico et al. [11] it was Lavauden [12] who first introduced the term desertification, and related it to poorly managed and low-productive rangelands in Tunisia. But it was Aubréville [13], a French forester, who introduced the term desertification in a broader sense. Aubréville [13] noticed forest degradation in West Africa as a result of destructive land use and adverse management. He used the term desertification to describe desert-like conditions as a result of land degradation in the West African tropical forest belt (700–1500 mm annual rainfall) [14].

During the relatively wet episode in the Sahel during the 1950s and 60s there was not much debate about desertification, but the discussion was revived during the drought of the early 1970's [10]. The international community became concerned about the problems of drought and environmental degradation in the Sahel. In 1977 the UN Conference on Desertification (UNCOD) was organized at Nairobi to discuss and formalize the desertification phenomenon. The UNCOD [15] described desertification as:

> " ... the diminution or destruction of the biological potential of the land, which can lead ultimately to desert-like conditions. It is an aspect of the widespread deterioration of ecosystems, and has diminished or destroyed the biological potential, i.e., plant and animal production, for multiple use purposes at a time when increased productivity is needed to support growing populations in quest of development."

The UNCOD resulted in an UN endorsed Plan of Action to Combat Desertification (PACD), which was coordinated by the UN Environmental Programme (UNEP). National PACDs were developed in the desertification-prone countries, but most of those plans were never carried out due to lack of funding [16]. Despite the initial definition of desertification provided by Aubréville [13], the focus was now on the arid, semi-arid, and dry sub-humid drylands, and not on the tropical forest zone that receives higher amounts of rainfall.

Many different definitions of desertification were proposed during the 1970s and 1980s, which led to confusion about what desertification actually is. To some, desertification meant a process of change, while others believed that desertification is an end-result of process of change. The common factor in all definitions is that desertification means an adverse environmental process [14], which results in long-lasting and possibly irreversible desert-like conditions [16]. Figures 1 and 2 illustrate such environmental changes for an area in Patagonia, where active wind erosion processes may result in a desert pavement that could be considered as a nearly irreversible desert-like end stage.

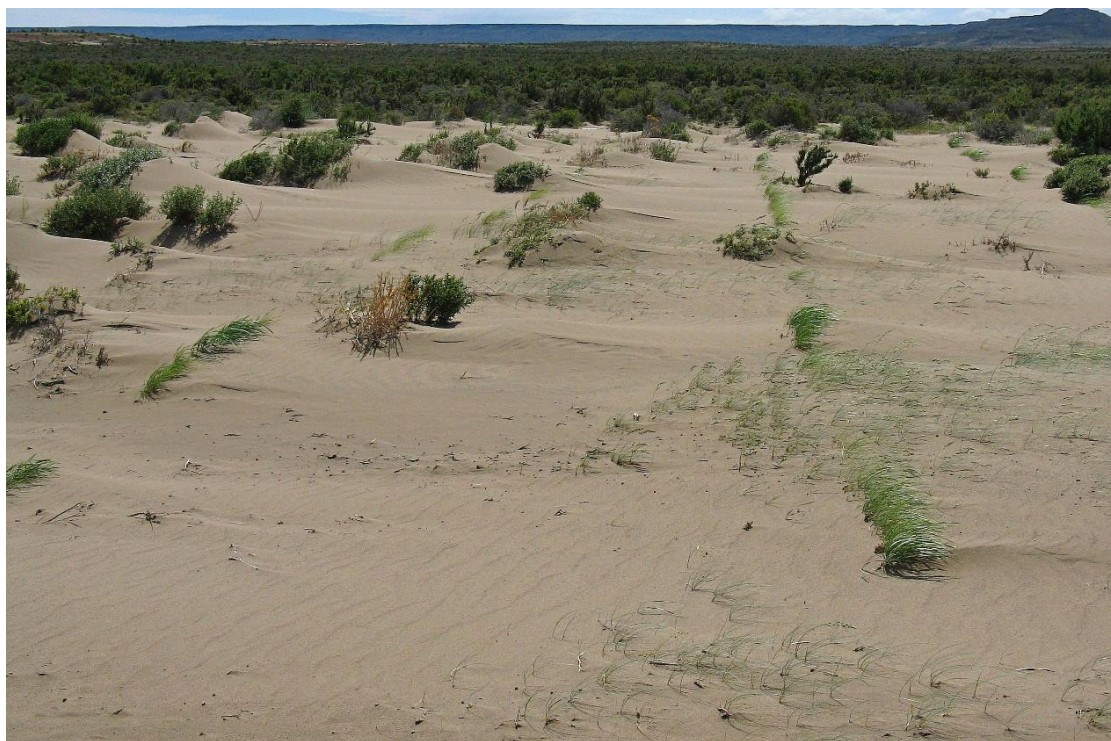

**Figure 1.** An example of a process of change: Active wind erosion destroying a vegetated area used for extensive sheep grazing in Chubut province, Patagonia, Argentina.

In 1991, UNEP defined desertification as land degradation in arid, semi-arid, and dry sub-humid areas resulting mainly from adverse human impact. This definition was again modified during the Earth Summit at Rio de Janeiro in 1992 into [17]:

> "Desertification is land degradation in arid, semi-arid and dry sub-humid areas resulting from various factors, including climatic variations and human activities."

This definition does not include the desert-like end stage, but merely describes desertification as land degradation in the drylands as a result of climatic impacts and human land management. It should be noted that land degradation includes soil degradation, as well as the degradation of vegetation resources. Hence, the definition considers desertification as a gradual process of soil productivity loss and/or a deterioration of vegetative resources because of human activities and climatic variations such as prolonged droughts and floods [18]. The climatic variations may also include climate change that may result in more severe drought or erratic rainfall patterns that impact the biological productivity of the land.

Many new definitions of desertification (e.g., by the Millennium Ecosystem Assessment) were developed since 1992, but in this article, the leading definition of desertification is the one defined in [17]. Following the Earth Summit, the UN Convention to Combat Desertification (UNCCD) was established in 1994. UNCCD's task is to improve the living conditions for people in drylands, to maintain and restore land and soil productivity, and to mitigate the effects of drought [19].

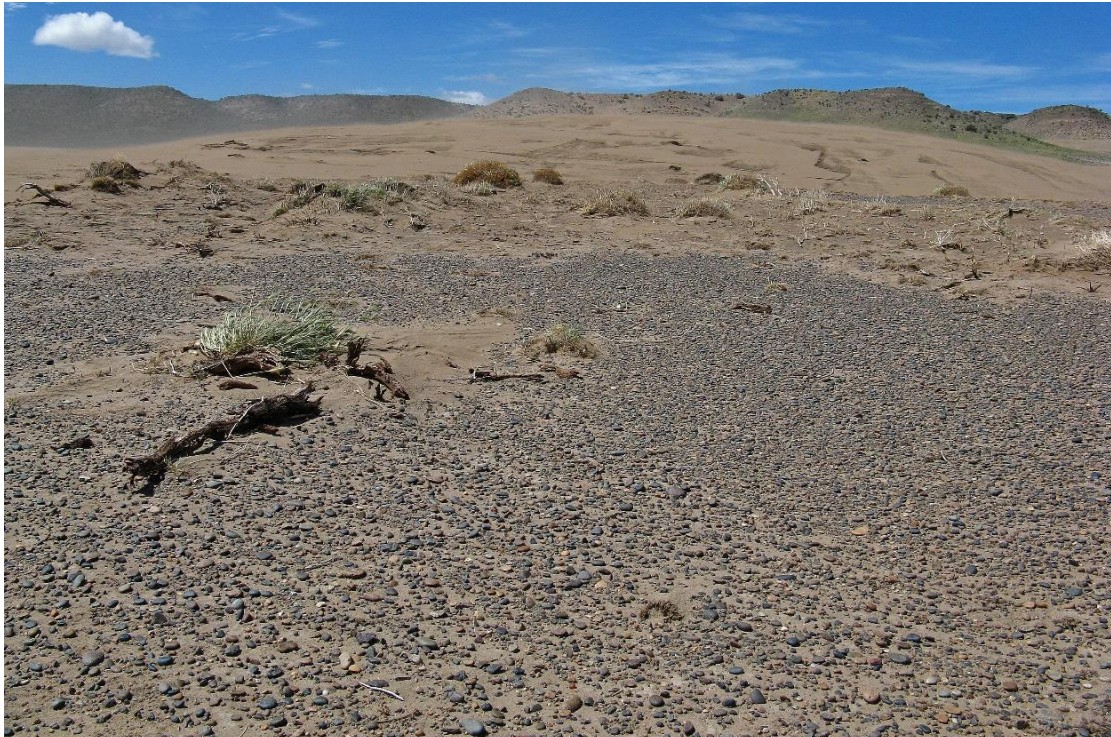

**Figure 2.** An example of a possible desert-like end stage: A desert pavement following strong wind erosion on previously vegetated land in Chubut Province, Patagonia, Argentina.

## 3. Global Assessment of Desertification

The earliest attempts to quantify desertification assessed the rate of Sahara desert encroachment [1,20]. Others criticized these assessments and found no evidence for large scale movement of the desert boundary but stated that the boundary between the Sahara desert and the more moist Sahelian zone is dynamic. That is, the alternation of dry and wet years results in commensurate movement of the apparent Saharan boundary, driven by rainfall anomalies [5,21,22]. But this view was in its turn criticized by other studies [23,24] which pointed at methodological flaws, and illustrated the controversy about desertification assessment.

Two global assessments of the status of desertification were conducted in the 1970s and 1980s (Table 1). The first assessment was made for the UNCOD in 1976–1977 [25] to produce a world desertification map [26]. The second assessment was done in 1983–1984 for UNEP to evaluate the progress in the implementation of the PACD [4]. The first (1976–1977) assessment estimated that an area of 3.97 billion ha, or 78.1% of the total drylands (excluding hyper arid deserts) was affected by desertification to some extent. The loss of productive capacity of the land resulted in an estimated annual economic loss of US $26 billion [25]. The second (1983–1984) assessment confirmed the seriousness of the problem and estimated an annual loss of land equal to 6 million ha to desertification. Another 21 million ha of land was believed to lose net economic productivity annually [4].

Although not intended as a desertification assessment per se, the World Map of the Status of Human-Induced Soil Degradation [27] was, and still is to some extent, a key product in the discussion on desertification. The map was developed by the Global Assessment of Soil Degradation (GLASOD) project and based on expert knowledge of soil degradation processes and their spread in a large number of countries. The map of human induced soil degradation distinguishes four categories of soil degradation (water erosion, wind erosion, chemical degradation, physical degradation), and shows the degree of these four categories at the global scale. Based on the GLASOD, the following numbers about soil degradation in drylands were estimated: 43 million ha of irrigated lands; 216 million ha rainfed croplands; and 777 million ha rangelands. In total, 1036 million ha (20.5%) of the total drylands

area was estimated as being prone to some form of soil degradation. As the definition of desertification also includes vegetation degradation, Dregne et al. [3] used the same GLASOD derived numbers and added another 2556 million ha of degraded rangelands (50% of the total drylands area) to the estimate of global dryland desertification. Hence, the assessment of Dregne et al. [3] concluded that 70.7% of the global drylands was affected by desertification. But by far the largest part was considered rangeland with significant vegetation degradation that was rather vaguely described as: "... e.g., all extensive areas of rangelands in Australia or the Aral-Caspian Basin of the USSR". No evidence for the massive rangeland degradation was provided.

**Table 1.** Reported assessments of the extent of desertification in the global drylands.

| Source | Remark | Area Affected by Desertification<br>ha | Proportion of Drylands *<br>% |
|---|---|---|---|
| UNEP (1991) [25] | Assessment for UNCOD, Nairobi, 1977 | $3.97 \times 10^9$ | 78.1 |
| UNEP (1984) [4] | Assessment for UNEP's Governing Counsel 12/9, 1984. | $3.48 \times 10^9$ | 68.4 |
| Mabutt (1984) [28] | Assessment of desertification seven years after UNCOD | $1.94 \times 10^9$ | 38.2 |
| Dregne (1986) [29] | Published study | $4.71 \times 10^9$ | 92.6 |
| Oldeman et al. (1991) [27] | Global Assessment of Soil Degradation (GLASOD). Assessment excludes rangeland degradation. | $1.04 \times 10^9$ | 20.5 |
| Dregne et al. (1991) [3] | Based on GLASOD numbers, with added rangeland degradation | $3.59 \times 10^9$ | 70.7 |
| Eswaran et al. (2001) [30] | Published study | $2.86 \times 10^9$ | 56.3 |
| Helldén and Tottrup (2008) [31] | Published study | None [#] | 0 |
| Bai et al. (2008) [32] | Published study | $0.38 \times 10^9$ | 7.6 |
| Le et al. (2014) [33] | Published study | $1.47 \times 10^9$ | 29.0 |

* Based on the estimated dryland area (excluding hyper-arid zones) of $5.08 \times 10^9$ ha by [34]. [#] Study identified overall greening in the global drylands and no signs of desertification were observed.

Several studies, as summarized in Helldén [5] and Verón et al. [35], have criticized the dramatic desertification numbers of the 1970s, 1980s, and early 1990s [3,4,25,27–29]. Research in Sudan neither found any evidence for desert encroachment nor for the establishment of permanent desert-like conditions around villages and water holes [5]. It has become clear that the estimate of 50% of degraded rangelands in the drylands by Dregne et al. [3] is meaningless as long as the desertification is not clearly specified or the method to measure it is not provided [35].

Desertification assessments since the year 2000 were largely based on available global databases of soils, rainfall, human population [30], or remote sensing of vegetation cover trends in drylands [31,32]. Eswaran et al. [30] combined the FAO/UNESCO Soil Map of the World with a global climate database containing data from 25,000 weather stations to compute soil moisture and temperature regimes. Nine classes of land quality (from class I —'very favorable for grain production' to class IX—'grain production virtually impossible') were calculated using 24 land stresses. The resulting map of vulnerability to desertification was overlain on a global population density map to determine the number of people affected by different degrees of desertification vulnerability. The results indicated that 14.6 million $km^2$ of land were at low risk, 13.6 million $km^2$ of land experienced moderate risk, 7.1 million $km^2$ were at high risk, and 7.9 million $km^2$ of land were under very high risk. The high and very high classes of vulnerability to desertification together covered 11.6% of the total land area, or 29.5% of the total drylands area, while 11.2% of the global population lived in these vulnerability zones. The assessment indicated that another 26.8% of the drylands was under moderate desertification vulnerability. Hence, the assessment of Eswaran et al. resulted in an estimated 56.3% of the total drylands area as threatened by desertification to some extent [30].

Satellite remote sensing imagery is available since the 1970s and provides a good option to assess Earth surface properties over time and at a vast spatial scale. Helldén and Tottrup [31] used the Normalized Difference Vegetation Index (NDVI), derived from the NOAA AVHRR dataset GIMMS (Global Inventory Modeling and Mapping Studies), to determine rates of desertification over the period 1981–2003. NDVI can be considered a proxy for Net Primary Productivity (NPP), and changes in NDVI can be interpreted as changes in ecosystem functioning and productivity [36]. The NDVI time series were standardized and compared with rainfall data for the same time period. The results showed that NDVI and rainfall are well correlated, and a general greening of the drylands was observed. The strongest greening occurred in the Sahel, followed by the Mediterranean and East Asia. The strong greening in the Sahel can be explained by the return of the rains to the level before the major drought period in the 1970s and early 1980s. The study did not find any signs of increased land degradation or desertification, as indicated by decreased vegetation cover over vast areas. This result questions the common assumption that desertification is spreading rapidly and is a serious threat to the livelihood of people in the drylands. But the coarse spatial resolution of the imagery (8 × 8 km) may have obscured the desertification happening at smaller, local scales.

A similar approach was followed by Bai et al. [32], who used the same GIMMS dataset to calculate NDVI time series, but also correlated the NDVI values to NPP values derived from MODIS. In addition, trends in rain use efficiency adjusted NDVI's were determined at the global scale. The results showed that there was an increase of 3.8% in the global NDVI (and NPP) over the period 1981–2003. The increase varied from 3% in Africa and North America to 6% in Asia. However, an increasing NDVI was not observed everywhere; 24% of the total land area suffered from a declining NDVI. When only drylands were considered, negative trends were observed in southern Africa, north-central Australia, and central-east Argentina. Overall, the reduced NDVI's were mainly in the humid regions (78%), while notorious environments like the Sahel actually showed an increase in NDVI. The authors of [32] cautioned for the use of NDVI to assess land degradation, and stated that NDVI only provides a proxy for land degradation, but does not include the effects of land use changes or show symptoms of land degradation processes like soil erosion, salinization, etc. For instance, a change from forest cover to crop land or grazing land results in a lower NDVI, but does not automatically mean that the land has become degraded. Still, it is concluded that the NDVI trends may highlight places where biological change is happening.

The NDVI maps of Bai et al. [32] were incorporated in the FAO Global Land Degradation Information System (GLADIS). Subsequently, Bai and Dent [37] evaluated the land degradation status map for South Africa by comparing point and areal field estimates with the map classes. They obtained 33% agreement for the point estimates, and 48% for the areal-based estimates. A number of problems were identified (for instance, climatic influence, crop lands being falsely classified as degraded), which Le et al. tried to correct [33]. This new global map of land degradation status was evaluated by comparing ground-based surveys of land degradation with the NDVI-based land degradation map [38]. Intermediate agreement between field based and remotely sensed based land degradation status was obtained. Similar problems as those discussed by Bai and Dent [37] were encountered. These problems are (i) the lack of observability of specific land degradation processes by coarse scale remote sensing imagery; (ii) fluctuations in vegetation cover in dynamic farming landscapes; (iii) the coarse resolution (~8 km) of the global remote sensing imagery prohibited the discrimination between improvements and degradation coexisting at smaller spatial scales.

## 4. Sahelian Desertification Based on Remote Sensing

Desertification research has been conducted in many dryland environments, such as China, Southern Africa, and Argentina, but the majority of desertification assessments have focused on the African Sahel, which generally is considered to be the global hotspot of desertification. Several NDVI-based studies observed greening of the Sahel between the mid 1980s and early 21st century when rainfall recovered to similar amounts as before the drought years [22,39–41]. Similar greening

trends were observed at other dryland regions [31] and were related to enhanced precipitation over the previous three decades [42]. These studies relied on the concept of Rain Use Efficiency (RUE), which is the ratio of Net Primary Production (NPP) and the rainfall [43]. It is usually assumed that land degradation reduces the RUE, because more rainfall gets lost as surface runoff and soil evaporation due to a degraded vegetation cover. In this way, reductions in RUE can be interpreted as indications for land degradation, independent of climate effects [44].

The use of RUE as a measure for desertification was criticized by Hein and De Ridder [45], who hypothesized that the increased precipitation and the higher RUE actually masks the human-induced land degradation, but this view was strongly rejected by Prince et al. [46] who mentioned several flaws in available datasets and the data analysis made by Hein and De Ridder [45]. In a new study, Hein et al. [47] re-analyzed part of the original data of Hein and De Ridder [45] and added new vegetation data from the Sahel. Their new analysis showed that the relation between the inter-annual variation in NPP and rainfall is non-linear for the Sahel. There are different relations for sites with varying rainfall amounts, which will influence the degree of impact land degradation may have on the remote sensing derived NPP – rainfall relation. Hein et al. [47] argued that more field data on vegetation species, densities, and productivity are required to meaningfully analyze remote sensing products on RUE and land degradation occurrence.

More recently, a study of vegetation changes in 260 Sahelian watersheds in Senegal, Mali and Niger was reported [48]. The study covered 30 years, from 1983 to 2012, and was based on the African Rainfall Climatology database [49] and AVHRR GIMMS NDVI data. The relation between rainfall and NPP was best described by a linear relationship in nearly half of the watersheds (128), while the other half of the watersheds (132) showed a non-linear relationship. It was concluded that the type of relationship has little impact on detecting greening or degradation trends. Overall the study showed again that nearly all watersheds experienced re-greening over the study period, with only four watersheds having experienced significant long-term declines in RUE [48].

A study covering entire West Africa [50] used remote sensing imagery to detect land use and land cover changes since 1975. It was shown that in the dryland part of West Africa major shifts of savannah land to agricultural crop land had occurred over the study period (1975–2013). Again, the overall re-greening of the Sahel was apparent, and also some hotspots of Farmer Managed Natural Re-greening (FMNR) were identified, especially in Niger and Burkina Faso. Hence, the remote sensing based studies leave little doubt for the re-greening of Sahelian Africa following the long drought of the 1970s and 1980s.

A problem with the NDVI-based assessments of desertification is the inability to determine vegetation species composition. A change from grassland to scrubland is often perceived as land degradation by local communities, but generally not recognized as such by scientists who base their analyses on remotely sensed data. Satellite-derived greening trends cannot always be interpreted as an improvement or recovery. In Senegal, Herrmann and Tappan [51] studied vegetation changes in the field and from remote sensing, and evaluated the perceptions of the local farmers. The NDVI based analysis showed greening of the area, but in the field a change in vegetation species was observed, which was interpreted as impoverishment by the local people. Generally, a shift in vegetation species and richness was observed. Tree coverage had declined and was replaced by more drought-tolerant shrub species. Similar changes in vegetation cover were reported from the Chihuahuan desert in the USA where grassland was replaced by desert scrub, which was believed to be caused by desertification. But the above ground biomass only marginally decreased [23]. Similar observations were made by Van Auken [52], who related the increased shrub cover on semi-arid grasslands in the southwestern part of the US to herbivore grazing. Grazing resulted in lowering of the grass biomass, which reduced the amount of fine fuel and an almost complete elimination of grassland fires. The lack of fire favored the encroachment of woody shrubs [52].

Higginbottom and Symeonakis [44] also concluded that the developed and applied NDVI-based methods lack validation of their results, which makes the large-scale analysis of desertification uncertain.

Like Hein et al. [47], they made a plea for the use of better field data and standardized validation methods of remote sensing based vegetation analyses. This should also include stakeholder perceptions on the type and use of vegetation in a specific area.

High-resolution, multi-spectral remote sensing data, processing techniques, and analysis methods have become widely available for many environmental monitoring applications [53]. However, for desertification assessments remote sensing can only provide indicators of land degradation, such as vegetation cover changes based on the NDVI, but cannot actually assess the degree of land degradation. Trends in those indicators, either negative or positive, do not provide immediate evidence for large scale degradation or improvement in ecosystem services [54]. At best, the remote sensing based analyses provide estimates of vegetation degradation, but do not provide reliable estimates of soil degradation processes that play a role in desertification. In-situ processes such as wind and water erosion can occur at spatial and temporal scales that do not match the characteristics of the remote sensing data [54]. Moreover, several soil degradation processes may occur in the same area [55], making detection even more complicated. For small-scale phenomena, such as field scale wind erosion or gully erosion, multi-spectral remote sensing imagery with a high spatial resolution (~1 m) is required, and actually becomes increasingly available [53,54]. But those images have a low temporal resolution, with the satellite passing only once every few days. It is likely that dust storms (Figure 3) or water erosion events are completely missed, but the high-resolution imagery can be used to asses gully erosion (Figure 4) rates and extent [56], as well as surface crusting [57]. Imagery with a low spatial resolution (>10 km) has a much higher temporal resolution, which can be used to detect large-scale dust storms [58], but is much too coarse to detect actual degradation processes on the ground.

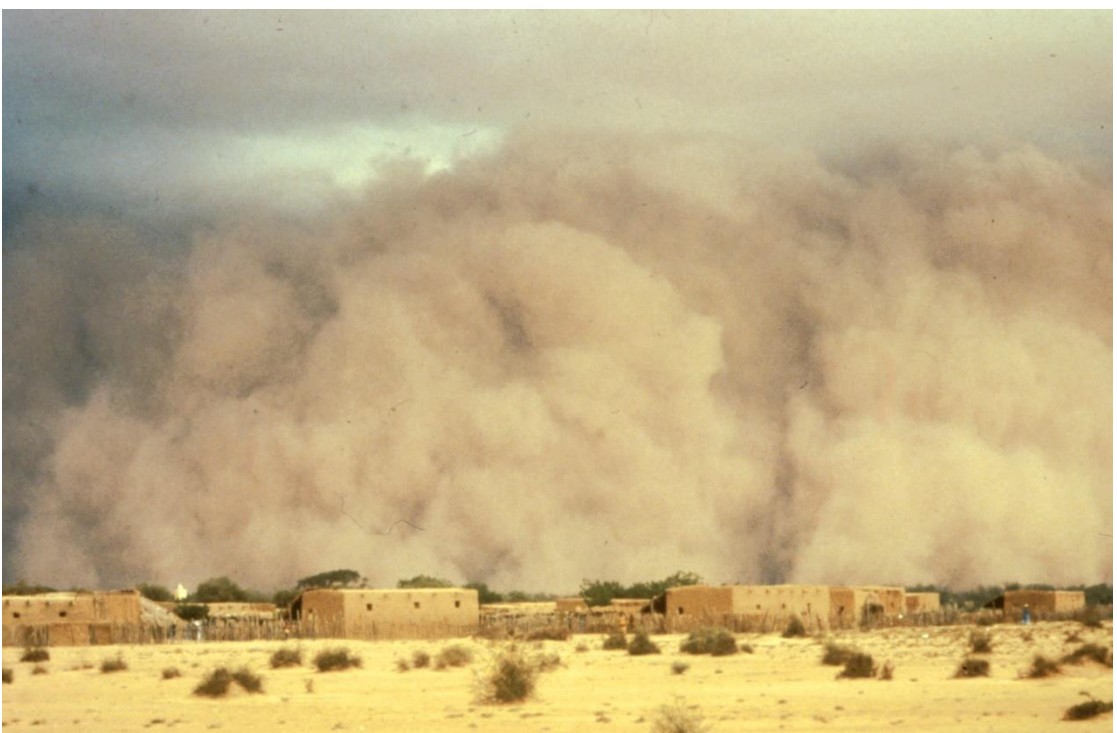

**Figure 3.** A dust storm near a village in Senegal.

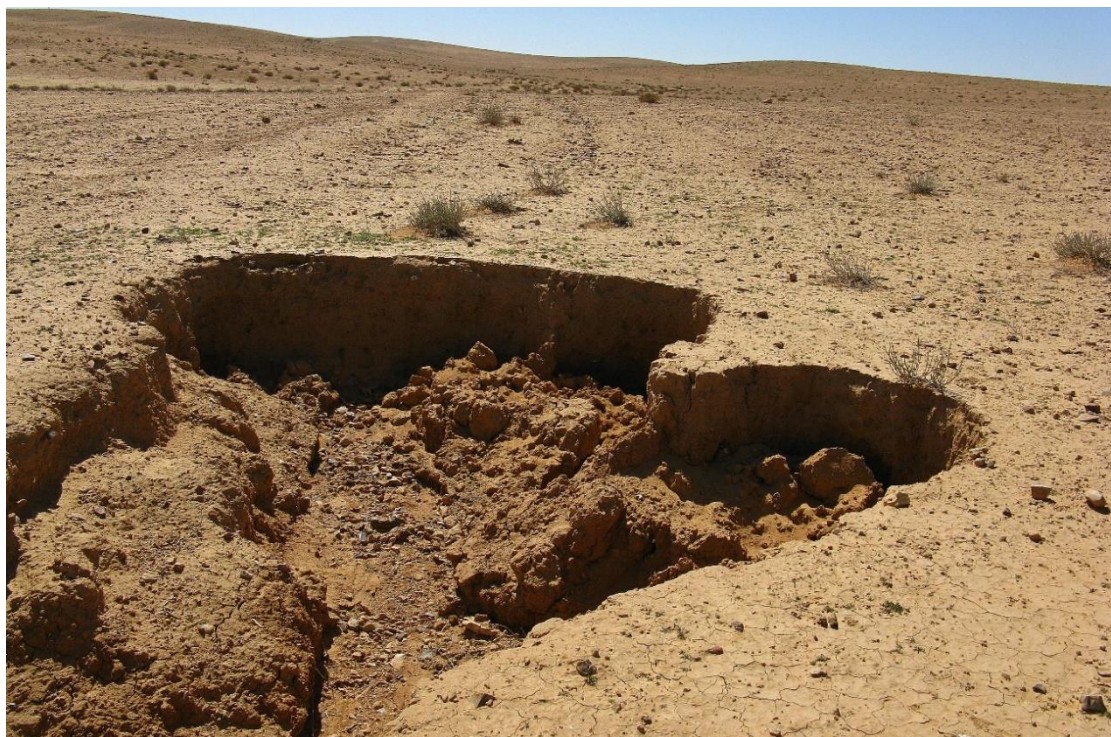

**Figure 4.** An active gully head eroding rangeland soil in the semi-arid 'Badia' of Jordan.

## 5. Soil Degradation Processes and Soil Quality

It is unlikely in the short term that better remote sensing based desertification assessments than the current ones will become available. So, the question "how severe is desertification in the global drylands?" remains largely unanswered in the scientific literature. Vegetation degradation data rely mostly on remotely sensed data, as described in the previous sections. Quantitative data of the actual soil degradation processes in terms of soil productivity losses in the drylands are scarce. For water erosion assessments, most studies have relied on modeling [59]. Such modeling studies show the global hot spots of water erosion, but do usually not include the actual loss of soil productivity. A meta-analysis of measured water erosion quantities [60] showed that most datasets come from more humid environments. The majority of the drylands are lacking such water erosion datasets, except for parts of the US and Mediterranean Europe. Concerning wind erosion data in the drylands, the situation is even worse. Many Aeolian studies have focused on geomorphological features such as dunes [61], but only a few studies dealt with quantified soil degradation by wind erosion in drylands. Most of this research was conducted in the US, China, and the Sahel, but the results are fragmented, based on poorly comparable research methods, and generally do not exceed the scale of a single agricultural field [62]. Hence, the in-situ quantitative scientific evidence for the degree of land degradation caused by wind and water erosion is meagre, which is also the case for other soil degradation processes.

Based on a review of many published articles on Sahelian land degradation, [63] concluded that the discrepancy between studies is related to the many definitions of land degradation and their different interpretations. These differences in definitions and interpretations can result in widely varying indicators of land degradation that may cause discrepancies in desertification evidence. The UNCCD defines land degradation as:

> "reduction or loss, in arid, semi-arid and dry sub-humid of the biological or economic productivity and complexity of rainfed cropland, irrigated cropland, or range, pasture, forest and woodlands."

This definition includes three different indicator categories: (i) biological productivity, (ii) economic productivity, and (iii) ecosystem complexity. Each of the three categories has its own indicators that

are difficult to define and quantify, which may explain the contradictory 'evidence' for Sahelian desertification. Rasmussen et al. [63] provided an example which illustrates the problem. When a semi-natural savannah grassland is converted to cultivated cropland, then the economic productivity is likely to increase, while the ecosystem complexity is declining. The biological productivity could either increase or decrease. Hence, land degradation assessments based either on in-situ observation or remote sensing may come to different conclusions, because different indicators will be used.

One alternative way to try to answer the question of the seriousness of desertification is to analyze soil quality changes in the drylands. If desertification has occurred for a long time in a certain area, the soil quality is likely to be reduced. Wind and water erosion will lead to a loss of soil, and preferential removal of the finest soil particles, organic matter, and nutrients [64,65]. This leads to less soil depth, lower water holding capacity due to soil structure decline, and reduced soil fertility. As a proxy for the degree of soil degradation, the soil organic matter (SOM) or soil organic carbon (SOC) content in the topsoil can be used as a measure of soil quality [66,67]. A literature search did result in only two studies that have quantified soil quality changes over time.

In a recent study, seven dryland locations were selected where soil sampling was carried out recently (after 1987) and in the past (before 1987) [68]. The data used in the study was available through the World Soil Information Service (WoSIS) [69] and Africa Soil Information Service (AFSIS) soil databases [70]. The average SOC contents in the topsoil at two moments in time were compared and it was evaluated if the area has undergone any degradation or improvement. Out of the seven studied locations, five locations showed an increase in SOC contents, and the SOC contents had only decreased in two locations. These changes were only significant ($\alpha < 0.05$) at two locations. In Mexico, at Mapimi, the average SOC content had decreased from 9.4 g/kg around 1980 to 2.6 g/kg in around 2002. The reason for this decline was most likely due to chemical deterioration of the soil as a result of soil mining activities. In Burkina Faso, the average SOC contents in the topsoil showed a significant increase from 6.7 to 8.5 g/kg over the period 1973–1994. This suggests that soil quality actually had improved in this part of the Sahel, which is at odds with the much acclaimed desertification problems in the Sahel. The increase in topsoil SOC contents in Burkina Faso were explained by the Sahelian re-greening, which has resulted in more organic material in the soils [68]. A similar result was obtained for an area in East Burkina Faso, where topsoil SOC contents had generally improved over a 27-year period (1969–1996) [66].

The seven areas studied by Van Vliet [68] are by no means sufficient to draw conclusions about the degree of global desertification. One of the main problems that limited the study to seven areas alone is the low availability of multi temporal soil data of the same locations. This lack of multi temporal data complicates the global desertification analysis, and can only be improved once more soil data become available in the WoSIS, AFSIS and other open databases. However, the analysis indicates again that in most instances, there is no scientific evidence for serious desertification. Popular statements like " ... extensive land and soil degradation still occurs all over the world and fertile soil resources are still rapidly depleted, reducing the potential for food production ... " [67]; referring to the UNCCD convention text and achievements and "The world's soils are rapidly deteriorating due to soil erosion, nutrient depletion, loss of soil organic carbon, soil sealing and other threats, ... " [71] seem to be ill-founded. In fact, the report 'Status of the World's Soil Resources' [72] states that it provides a baseline inventory of soil resources and the impact human actions and natural processes have on soils at a point in time (2015). New data on a variety of topics (soil erosion, soil organic matter, nutrient depletion, etc.) should be compared to the 2015 assessment to draw conclusions on changes in the global soil resources, including desertification assessments [72].

## 6. Desertification Scale and Sustainability

The previous assessments of desertification targeted the entire global drylands region, using the underlying assumption that desertification is a process happening at large spatial scales, such as a landscape or a whole province in a dryland country. Perhaps this idea was driven by the extreme wind

erosion in the Great Plains of North America and Canada during the 1930s, a period known as the Dust Bowl [73]. Years of cultivating and ploughing the prairie soils resulted in massive dust storms during the prolonged drought of the 1930′s. The wind erosion caused many families to give up farming and migrate to other areas, hoping to find work. But the lack of work for those newcomers resulted in the largest socio-economic tragedy in the USA and initiated research in soil erosion problems and soil conservation measures [74].

This example shows how a land degradation process in a dryland environment may have a large impact on the land and the people who depend on the same land for a living. But is it also a clear case of desertification? According to the UNCED [17] definition it is, but at the same time, the degradation did not lead to a loss of productive agricultural land. Since the 1930s, the Dust Bowl land has still been used for agriculture, albeit with soil conservation measures now in place [73]. The other issue is the spatial scale of the wind erosion problem itself. The dust storm may originate from a relatively small area of land, while the off-site impacts may affect a much larger area by reducing visibility, polluting air and surface water, and disrupting machinery, houses, etc. The actual damage to soils may in fact be limited to a relatively small portion of the total affected area. In fact, other soils that receive dust inputs may be enriched in organic matter and nutrients [75], which increases their productivity.

Land degradation, and thus desertification, is likely to occur mainly at fine spatial scale, such as a farmer's field [76], which makes it difficult to assess desertification at province, district, national, or continental levels. Even at the field scale, quantification of the actual amount of damage caused by land degradation processes such as water and wind erosion is difficult due to (i) the general short measurement periods, which may lead to too few or too many extreme events, (ii) measurement errors in the applied techniques, and (iii) spatial scale effects making generalization of results problematic [77]. Despite these uncertainties, many studies have published serious soil loss rates caused by wind erosion [78] and water erosion [79], nutrient losses [80], and surface crusting [81]. Generally, those studies present a bleak picture of many dryland environments, especially those where smallholder farmers grow crops and herd livestock using little inputs. This raises the question of how farmers have been able to cope with this severe degradation [77]. The answer to this question can be sought in the sustainability paradigm [82].

In the case of desertification, the sustainability paradigm considers land management in two mutual exclusive cases: strong sustainability versus weak sustainability [82]. If a farmer owns more than one field, he or she may not invest equally in each field, because of their limited natural, human, and economic capital. Hence, some fields may be well protected against degrading forces, while other fields may be left eroding or prone to other degrading processes. However, if no critical limit is passed, the previously degrading fields may be regenerated again when resources are better or more food production is needed. Conserving soil as much as possible, a case of strong sustainability, is generally promoted by soil scientists and agronomists. They see soil as a non-renewable natural resource, which is part of the natural capital a farmer owns. But a livelihood perspective sees soils as one of the many forms of capital, which can be partly exploited to achieve socio-economic sustainability [82]. This is a case of weak sustainability which allows the substitution of natural by human capital, as long as the total capital is maintained or enhanced [83]. Given that soil is only one form of capital needed to procure enough food, it can be a rational decision for a farmer to allow degradation on one or more fields. If the available household labor can produce more income by selling it elsewhere, which will allow buying food in the market, the required investments in sustainable crop production may not be made. But this type of decision may increase livelihood success [84]. Warren et al. [82] conclude that weak (or sensible) sustainability is a wise option that emphasizes the interdependence between natural resource conservation and the maintenance of socio-economic systems. Conservation of natural capital, such as soil, vegetation, and water resources, is part of a complex trade-off between agricultural and non-agricultural activities.

### 7. Policy View on Desertification

In the past years major institutions that are dealing with desertification problems have developed analysis tools, documents, and websites on global desertification. These new products make use of the latest global datasets on a variety of socio-economic and bio-physical factors that play a role in desertification. The most recent products that are briefly discussed here have been published by Conservation International, the European Union (EU), the UNCCD, and the Intergovernmental Science Policy Platform on Biodiversity and Ecosystem Services (IPBES).

Conservation International through its website Trends.Earth provides a platform for monitoring global land changes. It is intended to track achievement of the Sustainable Development Goals, especially SDG 15.3, by monitoring three indicators: land productivity, land cover, and soil organic carbon. The tool supports the mapping, monitoring, and reporting of land degradation and improvements, including the impacts of sustainable land management options. Countries can apply the tool to analyze data to prepare for their reporting commitments to the UNCCD.

The EU published the 3rd World Atlas of Desertification (WAD3) [85], which provides a valuable resource for global, regional, and country level assessments of a number of land degradation problems. It can assist policy-making and help to find environmental solutions. The WAD3 mapped and described 14 variables that play a role in global land degradation. The authors claim that no final assessment on the degree of desertification has been provided, as it was realized that " ... desertification or land degradation cannot be captured in global maps ... " Instead, the 14 identified variables were assessed at the global scale and diligently mapped. One of the mapped variables is the Land Productivity Dynamics (LDP), which is a measure of above ground biomass derived from high-resolution (1 km, 10 days) satellite imagery (SPOT-VGT). The global map shows areas with changes in LDP for the period 1999–2013. In total, 20.4% of the land surface (22 million ha) showed a persistent declining trend, with large differences between continents. Given the short period of analysis (15 years), it is not really possible to determine whether these changes are due to land degradation. Persistent drought, like the Millennium Drought in Australia, is also exemplified in the map. The authors of the WAD3 caution for conclusions on the numbers by stating: "Declining trends do not indicate land degradation per se, nor do increasing trends indicate recovery." However, the results of the analysis were already used before the publication of the WAD3 in the Global Land Outlook published by the UNCCD [6]. One of the key messages from the Outlook is "over the last two decades, approximately 20 per cent of the Earth's vegetated surface shows persistent declining trends in productivity, mainly as a result of land/water use and management practices." On their website, the UNCCD [86] makes an even stronger claim of the severity of desertification by stating "12 million hectares of productive land become barren every year due to desertification and drought alone", which seems to be based on the WAD3 land productivity dynamics estimate for the drylands.

In a special report on combating desertification in Europe [87] the degree and change in spatial extent of desertification was determined in 2008 and 2017. A dramatic increase of 177,000 km$^2$ in the area of high to very high desertification sensitivity was reported for southern Europe. This estimated increase was based on a methodology that uses three factors to calculate a desertification severity index. One of the factors is the Aridity Index, which is the ratio between annual rainfall and annual potential evapo-transpiration (PET). In the 2017 assessment the PET calculation used a different method, which generally resulted in a higher amount of PET [88], and thus an increase in the Aridity Index, which accounts for 1/3 in the desertification index. Hence, the dramatic increase reported is not an increase in desertification per se, but mainly an effect of a new calculation method. If the same PET calculation would have been used in 2008, a much larger area would have fallen under the high to very high desertification classes. Obviously, the increase in desertification would have been less dramatic. Still, the authors of the report recommended the European Commission to establish national action plans to fight desertification and achieve land degradation neutrality, which was adopted by the European Commission. This example shows how the development of any sound policy can be impeded by uncertainty, inconsistency, and possible exaggerations in the mapping of desertification.

Finally, the Intergovernmental Science-Policy Platform on Biodiversity and Ecosystem Services (IPBES) published a report on global land degradation [89]. Drylands and desertification are specifically addressed in the report, and the report discusses the difficulty of mapping desertification due to the complex interaction of different processes that are important. It is even acknowledged that desertification in the Sahel is localized and not sub-continental in scale. Despite this critical reflection on desertification, the report claims in its Summary for Policymakers that "Desertification currently affects more than 2.7 billion people and can contribute to migration". It is even mentioned that this claim is well established. Hence, according to IPBES the entire dryland population is affected by desertification, but it is entirely unclear which data this statement is actually based on. In a very dramatic media release, the IPBES warns that the well-being of 3.2 billion people is at risk due to global land degradation, which apparently includes the entire dryland population.

These examples illustrate the disconnection between policy and science when it comes to desertification [9]. Where many scientific studies over the last 20 years have downplayed the importance of desertification in drylands, the policy-oriented studies keep on claiming that desertification is expanding and threatening the livelihoods of many people in the drylands. Science has shown that dryland agro-ecosystems are highly complex systems that can rapidly change from high to low biological productivity, while the economic productivity is actually enhanced [63]. A lower biological productivity should not be interpreted as desertification though.

It has become clear that desertification is linked to this complexity and the associated changes that can occur due to drought or management, but is not necessarily an irreversible process that destroys land to the level that it is not available for food production or other ecosystem services [82]. It seems that the institutional policy level adopts desertification as the dramatic view of the complete loss of land, which started in the 1980s with the false notion of the expanding deserts, and was later replaced by the UNCCD definition of land degradation in drylands. More dialogue between science and policy could help to adjust the international efforts to halt desertification and to achieve the sustainable development goals in the global drylands.

## 8. Concluding Remarks

In the 1980s and 1990s desertification was perceived as a global environmental issue threatening the lives of millions of people living in the drylands. Most of the desertification discussions focused on the Sahel region where serious drought had affected the livelihoods of the local rural populations. First estimates of the severity of desertification were dramatic [3,4,25,28,29], but were later criticized, because no scientific evidence could be provided to support the early assessments. In fact, what was then perceived as Sahelian desertification was in reality the consequences of serious drought, which later disappeared when the rains got back to normal and a large scale re-greening happened [31,32]. Evidence for the massive degradation and abandonment of land in the drylands is not available. In science, desertification is now perceived as a more local-scale degradation problem (Figures 1–4) where land management plays a crucial role in either stimulating or decreasing degradation processes. Desertification is not necessarily an irreversible process; soil can be considered a form a natural capital that can be partly exploited to maintain socio-economic livelihoods [82].

The role of science in the global desertification debate remains crucial. The natural and social sciences should work together and provide evidence about the extent and seriousness of desertification processes in dryland environments, its causes, consequences and how the dryland populations are affected. Land degradation, and thus desertification, cannot be understood in isolation of other trends in society and nature [90]. The use of common definitions and broadly accepted research methods will be needed to provide meaningful information about desertification [35]. So far, the large-scale assessments have mainly focused on vegetation cover changes, which were used as a proxy for land degradation. More work should be done on mapping and classification of the severity of actual soil degradation processes occurring in drylands. This should include processes such as wind and water erosion, as well as salinization and soil crusting. This may lead to better and more objective global

assessments of the state and changes of drylands. The report 'Status of the World Soil Resources' [72] could become a key reference to which all observed states and changes could be compared for objective assessments of the degrees of soil degradation since 2015. This will help policy institutions to target areas for immediate action, and will assist in developing meaningful solutions to fight real and serious desertification problems.

Despite the critical reflection on desertification by scientists, the policy domain still considers desertification as a serious threat to the livelihoods of the global dryland populations. One can perceive the general dichotomy of science and policy from two different angles. From one perspective, science fails in providing hard and objective assessments and thus limiting informed decision making of policy makers. From the other perspective, it looks like policy makers tend to take numbers from the widely varying set of scientific reports that fit an alarmist message to stimulate action. A better communication between science and policy in which policy better facilitates the scientific assessments and where science listens better to the required insights is urgently needed to give global desertification the proper attention to deal with SDG 15, and specifically SDG 15.3, which aims at reaching land degradation neutrality (LDN) by 2030. LDN is a concept that was adopted by the UNCCD to maintain or improve ecosystem services, productivity, and resilience of the land [91]. It is a concept that reverses land degradation processes by implementing measures and strategies that are based on multi-variable assessments, considering land potential, land condition, resilience, social, cultural, and economic factors. Each country may have different land degradation problems, and may seek optimized solutions that fit the local circumstances. The design and implementation of such measures and strategies should be firmly based on scientific knowledge and policy objectives.

**Author Contributions:** Research idea and major part of the literature review: G.S.; review of soil quality issues: J.J.S.; writing of manuscript: G.S.; review and correction of draft manuscript: J.J.S. All authors have read and agreed to the published version of the manuscript.

**Funding:** This research received no external funding.

**Conflicts of Interest:** The authors declare no conflict of interest.

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
