# Peer review of "Desertification–Scientific Versus Political Realities"

_land, doi:10.3390/land9050156_

Round 1
Reviewer 1 Report
The manuscript reviews scientific versus political realities of the Desertification. Very nice and important work on the topic. The authors should consider the following concerns in improving the manuscript.
- Line 275: What about the spectral possibilities (i.e. number of bands) of sensors in such soil erosion analysis? Are they as critical as the pixel size or revisit time of satellites?
- Line 333: Burkina Faso, and 12 other countries in the Sahel are members of CILSS, a regional organization. The fight against the effects of drought and desertification is part of CILSS mandate. It has also responsibility to harmonize policies to increase food production in all country members. That includes soil conservation techniques. Is Burkina Faso an exception in getting SOC contents improved or it is a general trend in CILSS region?
- The strong re-greening in the Sahel at larger areas (e.g. lines 181,246,337,474) is under discussion for strong evidence as well. Maybe worth to check publications using moderate to high resolution satellite data for regional scales vegetation analysis. Evidence of large scale land degradation from remote sensing can be found in recent land cover / land use analysis in West Africa stating that “forests losses and natural habitat destruction are still causing severe water erosion, which removes the topsoil, reducing land productivity and creating conditions that lead to desertification”. All losses have been quantified (CILSS, 2016).
Cited document:
CILSS (2016). Landscapes of West Africa – A Window on a Changing World. U.S. Geological Survey EROS, 47914 252nd St, Garretson, SD 57030, UNITED STATES.
Reviewer 2 Report
The article presents a historical overview on the issue of desertification. Presenting definitions, scientific approaches used over time to assess it at both local and global scales, and presents the response of the international community to address it in the form of the United Nations Convention to Combat Desertification. The historical summary is very informative and relevant to present the evolution of our common understanding of land health, but the article reads more like an opinion piece than a true review article.
The article does a great job at identifying how the issue of desertification, as originally conceived, is much more limited in space and time than first assumed. The authors use that argument then to say that “The institutional policy level adopts desertification as the dramatic view of the complete loss of land, which started in the 1980’s with the false notion of the expanding deserts… It seems that strong claims on the extent and degree of desertification are made to justify the existence of policy institutions like the UNCCD, and maintain their international funding”. That statement fails to capture how the international policy bodies in the form of the UNCCD and the SDGs have evolved to incorporate the updated scientific knowledge we currently have and which the authors present so well in this article. As a minor note, I agree as many of us do, that the name of the convention (UNCCD) is now date and does not accurately reflect its objectives, but that is a naming issue, nothing else. Since the article is discussing the science and the politics of desertification, extensive sections “land degradation” and “land degradation neutrality” need to be added, since our current scientific understanding has proven that desertification at the global scale is not as serious as we first considered, but land degradation is, and LDN is the current political framework being used to address it. LDN address many of the issues the authors flag:
- It is much more than just vegetation (the indicators include changes in vegetation productivity, land cover, and organic carbon as a minimum, but also gives the option to countries to add their own set of indicators to reflect local condition and processes.
- LDN is reported at national scales, but the assessment is made looking at changes withing ecologically similar units, acknowledging that degradation is very context and system specific.
- Land degradation is defined, and I am paraphrasing, as a reduction in the capacity of the land to provide ecosystem goods and services which can be avoided, reduced, and reversed, which addresses one of the main issues the authors flag with desertification, how one directional that concept was. The international community, both scientist and policy makers, has acknowledge one of the main issues the authors flag. Not including that in the review is greatly misleading.
Reviewer 3 Report
62-111: History of desertification. Suggest this section is unnecessary for two reasons: (1) it seems every paper on the topic of desertification feels compelled to cite these exactwords, exact phrases, and tired truisms. Not sure how Figures 1,2 add to paper; and (2) most importantly, this inconsequential text detracts from the actual topic of the paper (hopefully, the disconnect between science and policy as stated in the Abstract).
91-93: Ironically, germane to putting this review in context is what seems to be the author’s choice of a definition of desertification, which from this statement: “The common factor in all definitions is that desertification means an adverse environmental process [14], which results in long-lasting and possibly irreversible desert-like conditions [16]”. This suggests an “end-point” of “desert-like” conditions. This is explicitly reinforced by their inclusion of Figs. 1-2. Furthermore, their view largely ignores what factors may be responsible for observed changes (at least in a systematic way). After all, “desertification” has been (and is) being “measured” by many factors, including NDV, grass cover, land cover, population density, patch-size distributions of vegetation, primary productivity, poverty, and so on. Furthermore, there’s another suite of studies that use a variety of climate change indices to map “desertification”. It’s not clear how this particular endpoint, “desert-like” conditions, fits into any of this. The authors write “For instance, a change from forest cover to crop land or grazing land results in a lower NDVI, but does not automatically mean that the land has become degraded” (lines 201-202). Hard to reconcile with desert-like conditions.
104-111: This review ignores what most readers would consider the most relevant/current definition of desertification/land degradation by the Millennium Ecosystem Assessment: “Desertification is a result of a long-term failure to balance demand for and supply of ecosystem services in drylands” (Millennium Ecosystem Assessment, 2005. Ecosystems and Human Well-being: Desertification Synthesis. World Resources Institute, Washington, DC). This seems especially relevant to this review since the authors cite the Sustainable Development Goal 15, “which includes combatting desertification as an important target for the sustainable use of terrestrial agro-ecosystems.” Also, in reviewing Hellden & Tottrup’s use of NDVI as a proxy for desertification (lines 177-178), the authors write: “NDVI can be considered a proxy for Net Primary Productivity (NPP), and changes in NDVI can be interpreted as changes in ecosystem functioning and productivity [34].” AND, furthermore, state that since Hellden & Tottrup did not find “any signs of any signs of increased land degradation or desertification, as indicated by decreased vegetation cover over vast areas” (i.e., via NDVI) (lines 91-93). This seems more in line with the MEA definition of loss of ecosystem functioning and productivity and is not consistent with the author’s emphasis of desertification as desert-like conditions (Figs 1-2)?
113-114: Suggest delete first sentence. Doing so will result in no loss of information. The sentence that follows has nothing to do with global scale. There are other issues in this sentence (!), so suggest delete.
114-115: Suggest change “tried to assess” to “assessed” and “[e.g., 1,20] to “[1,20]. (Incidentally, overall this paper is well-written but the writing is a bit rough in places and needs some editorial assistance.)
114-115: Suggest change “The alternation of dry and wet years results in an alternating movement” to “That is, the alternation of dry and wet years results in commensurate movement”
123 (numerous issues): Table 1: Reported assessments of desertification in global drylands. Lines 173-216 Various studies dealing with global assessments are described here but for some reason are excluded from Table 1. Why are discussed but not in table? Other assessments along the same line as those in the table include Bai et al., in paper entitled “Proxy global assessment of land degradation”(cited by the authors) who used remote sensing to conclude that 24–29% of global land area was experiencing decreases in biomass productivity between the 1980-2000s. Also, Goldewijk et al. (2017: Anthropogenic land use estimates for the Holocene – HYDE 3.2. Earth Syst. Sci. Data, 9, 927–953) argued that less than 10% of drylands are undergoing desertification but is occurring in those regions where 21% of most dryland populations exist. Lines 407-466 While this table proports to list assessments – at least up to 2001 – more recent ones are discussed in Section 7-Policy view on desertification
131-132: “The loss of productive capacity of the land resulted in an estimated annual economic loss of US $ 26 billion [3].“ Not clear if this statement refers to the first or secondassessment? Also, does this refer to the first assessment? 1976-1977 yet citation #3 is 1991. Is this correct? A bit confusing.
135-152: This section is somewhat unclear. In the abstract it is stated: “Many of the dramatic global assessments of desertification were heavily criticized by scientists working in drylands. The used methodologies, which were largely based on remote sensing data analysis, and the lack of ground-based evidence gave rise to critical reflections on desertification.” Yet GLASOD, which was a flawed, qualitative approach, is presented here as “a key product in the discussion on desertification.” Must disagree with the implication that since GLASOD “was a least based on some degree of evidence” that it is somehow an important product per se in the context of the goal of this paper. Maybe I’m misunderstanding the logic.
145: GLASOD is the subject topic of this text but then abruptly, the work of Dregne et al is introduced. Why? What’s the relationship between GLASOD and Dregne et al work? Not explained so hard to follow lines 148-152 in terms of their context.
153-157: What “dramatic desertification numbers” are being referenced here? UNEP? Oldeman? GLASOD? Dregne? All of Table 1 or only Dregne [ref 3]?
155-157: Criticize Dregne et al. [citation 3] here because “meaningless as long as the desertification is not clearly specified or the method to measure it is not provided” but what about GLOSOD? This is not clear since GLOSOD and Dregne combined here in paper.
158: “More recent” – what is recent? Based on citations, seems up to 2008. Ambiguous.
166-172: Verb tense not consistent with prior text in paragraph. In fact, this needs attention throughout text to check for inconsistencies.
188-196: This paragraph introduces LADA, which is specifically a dryland assessment program. Hence, confused why this sentence suggests otherwise: “When only drylands are considered….”
218: Sahelian desertification. This section is more about remote sensing. Based on the abstract, it’s hard to understand the rationale for an emphasis throughout this paper on case studies of the Sahel. Nothing contained here is new or original.
228: Who is “They”?
284-285: At this point in the paper, this statement seems very odd: “Measured data of land degradation processes in the drylands are scarce.” The reader must infer that this refers to soil quality (section heading)? But if so, why is soil quality now being highlighted? Up to here, no specific types of data that indicates “land degradation” seems to be highlighted. Am I again missing the logic?
467-497: The Abstract ends with the statement “This highlights the disconnection between science and policy, and there is an urgent need for better dialogue in order to achieve SDG 15.” I interpreted this as the goal of this paper. However, these concluding remarks are general (i.e., vague) and are simply a longer re-statement of this disconnect, not reaching any logical conclusions based on previous sections.
Reviewer 4 Report
Review of: “Desertification – Scientific versus political realities”
The manuscript provides an analysis of the phenomenon of desertification and its possible (or impossible) global assessment.
The authors highlight some of the practical issues associated with the assessment of desertification. Quite properly they highlight how the ongoing phenomenon of greening of dryland regions can be interpreted by looking at satellite imagery as a form desertification reversal, while an analysis son the ground can show that greening can indeed result from bush encroachment, a phenomenon likely associated with warming, increased atmospheric CO2 and land mismanagement that is a form of land degradation (see for instance Saha et al., 2015 for a critical discussion of this phenomenon and of the ongoing greening of regions affected by decreasing precipitation trends). At the same time, they show that there is often some confusion between the mapping of desertified regions and the mapping of drylands. I agree with both assessments but I also notice that these authors have also fallen in other forms of confusion. As will discuss below part of the confusion arises from a narrow definition of desertification adopted by these authors, which basically coincides with a definition of land degradation, to the point that in the manuscript these two terms are used interchangeably. Another more general criticism is that the authors refer to a very limited number of studies an, even worse, they claim that most studies concentrated on the Sahel, thereby ignoring a vast body of literature on Southern Africa, Australia, Central Asia, and the Americas.
Overall the manuscript can become an important addition to the literature after major revisions. In this review I refer to the papers I am familiar with them but you can find several other studies making similar points.
- I think it is important to define desertification in a broader sense than land degradation. The commonly adopted definitions include also climate change as one of the drivers in addition to land degradation. Drought can lead to desertification if desertification is defined as the transition to desert-like conditions. It is important to define this process as induced bu climate change or human action. It is also important to stress that the common notion of desertification is as a rather irreversible process in the sense that if drought is followed by a wet period in which the landscape recovers (e.g., grasses grow back), then we did not see a desertification. But if the loss of vegetation cover after the drought leads to either changes in climate or soil productivity, then we can say that desertification occurred. Positive feedback mechanisms that can “lock” the system in a “desert state” (e.g., Anderies et al., 2002; Okin et al., 2009). Land degradation feedbacks are only some of the processes. The comment made by the authors about productivity not being necessarily a good metric of desertification is a good one. This should be highlighted in a more critical analysis of the definition of desertification. A recent review chapter has clarified some of these points as well as the ambiguity of the notion of desertification itself “…These socio-environmental changes are often termed “desertification”, which literally means “transition to desert-like conditions”. Because deserts occur in different forms and shapes, desertification is a rather ambiguous term that does not uniquely point to a specific environmental process, final outcome, or underlying driver of environmental change in drylands. Indeed, desertification can lead to, among other outcomes, a loss of vegetation cover, accelerated soil erosion, bush encroachment in xeric grasslands, displacement of perennial native grasses by exotic annuals, and an increase in soil salinity or toxicity (e.g., Martinez-Beltrán and Manzur, 2005; Rengasamy, 2006; Schlesinger et al., 1990; Van Auken, 2000). These changes are relatively irreversible at time scales of a human generations. In other words, desertification is the environmental “deterioration” of drylands (Dregne, 1977) that leads to a persistent and, in the most extreme cases, even irreversible transition to desert-like conditions (MEA, 2005; D’Odorico and Ravi, 2016).” (“Dryland Ecohydrology”, Chapter 22 Springer, 2020). The reviewer does not expect the authors to agree on this point; but it is necessary to at least explain why they want to narrowly define the process as a mere land degradation. Admittedly, the manuscript does not focus on processes but on mapping (using land cover studies) and quantification and critical analysis of policy and political implication of global assessments of desertification (or lack of thereof). If anything, I would move to the very beginning the text you have now on lines 301-316 and broaden it ton desertification and not only land degradation.
- One of the major points made by the authors is that there are major limitations to global assessments of desertification, that most of them are decades old, and the more recent use of satellite data (e.g., NDVI) needs to be done with some caution because it does not account for shifts in plant community composition (e.g., bush encroachment) and therefore greening and browning analyses need to be combined with an analysis of the associated land degradation processes. It is important to add here ecological processes of changes in plant community composition as well climate related processes associated with land-atmosphere feedback. There is a rich literature on the way loss of vegetation cover may affect precipitation (See Nicholson, “Dryland Climatology”, Cambridge University press; Bonan, “Ecological Climatology”, Cambridge University Press). It is also important at this point of the manuscript (Section 3) to highlight the new Atlas of desertification that the authors mention at the end of the manuscript. The new atlas looks at desertification from multiple perspective, including salinization, land abandonment, livelihoods. It cannot be ignored. Likewise, initiatives to document and map soil health (e.g., “trends.earth”) need to be recognized along with their strengths and limitations
- The case of Sahel has been well-documented but it is important to recognize that many other regions of the world have been investigated and that the focus of this paper is strongly on remote sensing. Referring to shrub encroachment here and saying that it exists also in the Chihuahuan desert can be misleading because it does not stress that this phenomenon has been happening worldwide (see Van Auken, 2000; Archer, 1989). It is also confusing why the authors are a bit obsessed with NDVI and remote sensing, particularly when they state that wind and water erosion events can be missed by remote sensing. This part of the manuscript seems to be a critic to the use of remote sensing instead of an analysis of environmental phenomena. Many studies of wind erosion are indeed based on remote sensing of dust plumes at the regional scale. Regardless, it seems to me that this section makes a few random statements based on a limited analysis of the literature.
- Section 5 keeps mixing the terms land degradation and desertification. The authors review water erosion data sets and state that most water erosion studies were conducted in the Sahel. This is definitely not true! In fact, there have been decades of research (field research and model) developed in the USA and most of the leading models that are now applied worldwide were developed by the USDA. But there has been research also in Australia, China, and other regions of the world. Some of these statements perhaps reveal a limited familiarity with this literature. There is also some misconception here between the availability of datasets from a specific region and the fact that a certain process has not been studied in that region. Fundamental research contributed to the understanding of underlying processes. This is possible even without having mapped every square meter of the landscape.
- It is also not true that wind erosion is understudied and that “only few studies dealt with wind-driven land degradation of drylands”. There have been many decades of research in this field by a very active community of scientists. Please read the review by Ravi et al., (2011) for a comprehensive synthesis of the main findings.
- On lines 317-354 you provide a very scattered and incomplete analysis of the literature to then draw the rushed conclusion that seven studies are not sufficient to draw more general conclusions. You either engage in a comprehensive review on this topic or refrain from making these bold statements because you expose yourself to serious criticism.
- Section 6. You need to explain better the ‘sustainability paradigm’ you are referring to. This section seems to present your own personal opinion (and its is not clear what point you are trying to make) and needs to be anchored better to evidence from the literature otherwise it sounds speculative.
- Section 7: you critique of WAD3 is strongly affected by a narrow interpretation of desertification as land degradation and by a focus on the need to map desertified areas instead of looking at the global distribution of possible drivers and impacts (e.g., WAD3 gives maps of animal husbandry, salinity, water use, etc., including productivity), which in my view sounds quiet interesting. It is not clear what point you are trying to make and you keep jumping from point to point without a clear line of argument, with very incomplete analysis of the literature, criticizing WAD3 or the way it was used before it was even published. Your readers will be lost at this point. Are you again concerned about the fact that productivity is not a good metric to evaluate land degradation? Likewise, the comment on PET and changes in the way it was calculated in the estimation of the Aridity Index is a potentially interesting one: You should explain better and relate to the point you were trying to make in the previous and subsequent paragraphs. Perhaps the whole section should be about how the development of any sound policy is often impeded by uncertainty, inconsistency, an possible exaggerations in the mapping of desertification. How the paper has shown that part of them originate from ambiguities in the definition of desertification, etc.
- The statement you make at the end of section 7 that exaggerations in the estimates of desertification are done on purpose to justify the existence of agencies such at UNCCD is strong, subjective, poorly justified and does not belong to a research article. A statement like this one needs to be more nuanced. You could let the readers draw their own conclusions based on the points you have presented in your paper .
I hope these comments are useful
Minor points:
Line 40: “by scientists”
Line 39: delete “by”
Line 71: “describe the emergence of desert-like…”
Line 95: how do you know that it is irreversible?
Author Response
Your detailed report on our manuscript has arrived after we had made the requested changes by three other reviewers. Some of the comments were also mentioned by others and have been included, but unfortunately due a lack of time we could not fully include your remarks.
Round 2
Reviewer 2 Report
The authors have addressed the comments made by the three reviewers from the first round improving the manuscript following suggestions and explaining the reasons for which some of the suggested changes were not implemented. I don't have any further comments at this point.
Author Response
The reviewer has not made new comments, so nothing to reply to.
Reviewer 3 Report
None
Author Response

(The authors gave the same response as above.)

Reviewer 4 Report
My review was submitted with a delay and the authors did not have a chan ce to see it before resubmitting.
I see that some of the issues i have detected in the first submission are still there. I am pasting here my comments, in case they can be useful to the authors in their second revision. some of these comments may not apply anymore.
Of note is the confusion I saw between desertification and land degradation; the claims that most erosion studies focused on water and not on wind , and on the Sahel.
----------------
Review of: “Desertification – Scientific versus political realities”
The manuscript provides an analysis of the phenomenon of desertification and its possible (or impossible) global assessment.
The authors highlight some of the practical issues associated with the assessment of desertification. Quite properly they highlight how the ongoing phenomenon of greening of dryland regions can be interpreted by looking at satellite imagery as a form desertification reversal, while an analys son the ground can show that greening can indeed result from bush encroachment, a phenomenon likely associated with warming, increased atmospheric CO2 and land mismanagement that is a form of land degradation (see for instance Saha et al., 2015 for a critical discussion of this phenomenon and of the ongoing greening of regions affected by decreasing precipitation trends). At the same time, they show that there is often some confusion between the mapping of desertified regions and the mapping of drylands. I agree with both assessments but I also notice that these authors have also fallen in other forms of confusion. As will discuss below part of the confusion arises from a narrow definition of desertification adopted by these authors, which basically coincides with a definition of land degradation, to the point that in the manuscript these two terms are used interchangeably. Another more general criticism is that the authors refer to a very limited number of studies an, even worse, they claim that most studies concentrated on the Sahel, thereby ignoring a vast body of literature on Southern Africa, Australia, Central Asia, and the Americas.
Overall the manuscript can become an important addition to the literature after major revisions.
- I think it is important to define desertification in a broader sense than land degradation. The commonly adopted definitions include also climate change as one of the drivers in addition to land degradation. Drought can lead to desertification if desertification is defined as the transition to desert-like conditions. It is important to define this process as induced bu climate change or human action. It is also important to stress that the common notion of desertification is as a rather irreversible process in the sense that if drought is followed by a wet period in which the landscape recovers (e.g., grasses grow back), then we did not see a desertification. But if the loss of vegetation cover after the drought leads to either changes in climate or soil productivity, then we can say that desertification occurred. The review by D’Odorico et al (2013) positive feedback mechanisms that can “lock” the system in a “desert state” (e.g., Anderies et al., 2002; Okin et al., 2009). Land degradation feedbacks are only some of the processes. The comment made by the authors hof productivity not being necessarily a good metric of desertification is a good one. This should be highlighted in a more critical analysis of the definition of desertification. A recent review chapter has clarified some of these points as well as the ambiguity of the notion of desertification itself “…These socio-environmental changes are often termed “desertification”, which literally means “transition to desert-like conditions”. Because deserts occur in different forms and shapes, desertification is a rather ambiguous term that does not uniquely point to a specific environmental process, final outcome, or underlying driver of environmental change in drylands. Indeed, desertification can lead to, among other outcomes, a loss of vegetation cover, accelerated soil erosion, bush encroachment in xeric grasslands, displacement of perennial native grasses by exotic annuals, and an increase in soil salinity or toxicity (e.g., Martinez-Beltrán and Manzur, 2005; Rengasamy, 2006; Schlesinger et al., 1990; Van Auken, 2000). These changes are relatively irreversible at time scales of a human generations. In other words, desertification is the environmental “deterioration” of drylands (Dregne, 1977) that leads to a persistent and, in the most extreme cases, even irreversible transition to desert-like conditions ...” . The reviewer does not expect the authors to agree on this point; but it is necessary to at least explain why they want to narrowly define the process as a mere land degradation. Admittedly, the manuscript does not focus on processes but on mapping (using land cover studies) and quantification and critical analysis of policy and political implication of global assessments of desertification (or lack of thereof). If anything, I would move to the very beginning the text you have now on lines 301-316 and broaden it ton desertification and not only land degradation.
- One of the major points made by the authors is that there are major limitations to global assessments of desertification, that most of them are decades old, and the more recent use of satellite data (e.g., NDVI) needs to be done with some caution because it does not account for shifts in plant community composition (e.g., bush encroachment) and therefore greening and browning analyses need to be combined with an analysis of the associated land degradation processes. It is important to add here ecological processes of changes in plant community composition as well climate related processes associated with land-atmosphere feedback. There is a rich literature on the way loss of vegetation cover may affect precipitation (See Nicholson, “Dryland Climatology”, Cambridge University press; Bonan, “Ecological Climatology”, Cambridge University Press). It is also important at this point of the manuscript (Section 3) to highlight the new Atlas of desertification that the authors mention at the end of the manuscript. The new atlas looks at desertification from multiple perspective, including salinization, land abandonment, livelihoods. It cannot be ignored. Likewise, initiatives to document and map soil health (e.g., “trends.earth”) need to be recognized along with their strengths and limitations
- The case of Sahel has been well-documented but it is important to recognize that many other regions of the world have been investigated and that the focus of this paper is strongly on remote sensing. Referring to shrub encroachment here and saying that it exists also in the Chihuahuan desert can be misleading because it does not stress that this phenomenon has been happening worldwide (see Van Auken, 2000; Archer, 1989). It is also confusing why the authors are a bit obsessed with NDVI and remote sensing, particularly when they state that wind and water erosion events can be missed by remote sensing. This part of the manuscript seems to be a critic to the use of remote sensing instead of an analysis of environmental phenomena. Many studies of wind erosion are indeed based on remote sensing of dust plumes at the regional scale. Regardless, it seems to me that this section makes a few random statements based on a limited analysis of the literature.
- Section 5 keeps mixing the terms land degradation and desertification. The authors review water erosion data sets and state that most water erosion studies were conducted in the Sahel. This is definitely not true! In fact, there have been decades of research (field research and model) developed in the USA and most of the leading models that are now applied worldwide were developed by the USDA. But there has been research also in Australia, China, and other regions of the world. Some of these statements perhaps reveal a limited familiarity with this literature. There is also some misconception here between the availability of datasets from a specific region and the fact that a certain process has not been studied in that region. Fundamental research contributed to the understanding of underlying processes. This is possible even without having mapped every square meter of the landscape.
- It is also not true that wind erosion is understudied and that “only few studies dealt with wind-driven land degradation of drylands”. There have been many decades of research in this field by a very active community of scientists. Please read the review by Ravi et al., (2011) for a comprehensive synthesis of the main findings.
- On lines 317-354 you provide a very scattered and incomplete analysis of the literature to then draw the rushed conclusion that seven studies are not sufficient to draw more general conclusions. You either engage in a comprehensive review on this topic or refrain from making these bold statements because you expose yourself to serious criticism.
- Section 6. You need to explain better the ‘sustainability paradigm’ you are referring to. This section seems to present your own personal opinion (and its is not clear what point you are trying to make) and needs to be anchored better to evidence from the literature otherwise it sounds speculative.
- Section 7: you critique of WAD3 is strongly affected by a narrow interpretation of desertification as land degradation and by a focus on the need to map desertified areas instead of looking at the global distribution of possible drivers and impacts (e.g., WAD3 gives maps of animal husbandry, salinity, water use, etc., including productivity), which in my view sounds quiet interesting. It is not clear what point you are trying to make and you keep jumping from point to point without a clear line of argument, with very incomplete analysis of the literature, criticizing WAD3 or the way it was used before it was even published. Your readers will be lost at this point. Are you again concerned about the fact that productivity is not a good metric to evaluate land degradation? Likewise, the comment on PET and changes in the way it was calculated in the estimation of the Aridity Index is a potentially interesting one: You should explain better and relate to the point you were trying to make in the previous and subsequent paragraphs. Perhaps the whole section should be about how the development of any sound policy is often impeded by uncertainty, inconsistency, an possible exaggerations in the mapping of desertification. How the paper has shown that part of them originate from ambiguities in the definition of desertification, etc.
- The statement you make at the end of section 7 that exaggerations in the estimates of desertification are done on purpose to justify the existence of agencies such at UNCCD is strong, subjective, poorly justified and does not belong to a research article. I am not trying to defend UNCCD and, frankly, I always wondered what its mission is. But a statement like this one needs to be more nuanced. You could let the readers draw their own conclusions based on the points you have presented in your paper .
I hope these comments are useful
Minor points:
Line 40: “by scientists”
Line 39: delete “by”
Line 71: “describe the emergence of desert-like…”
Line 95: how do you know that it is irreversible?
References
Anderies, J. M., et al. (2002), Grazing management, resilience, and the dynamics of a fire‐driven rangeland system, Ecosystems, 5(1), 23– 44.
Archer, S. (1989), Have southern Texas savannas been converted to woodlands in recent history? Am. Nat., 134(4), 545– 561.
Bonan, 2008; “Ecological Climatology”, Cambridge University Press
Nicholson, “Dryland Climatology”, Cambridge University press
Van Auken, O. W. (2000), Shrub invasions of North American semiarid grasslands, Annu. Rev. Ecol. Syst., 31, 197– 215.
